# Social Insurance Burden and Corporate Environmental Performance: Evidence from China

**Nai-Chun Wang [1], Deng-Kui Si [1,\*] and Chun-Feng Dong [2]**

1   School of Economics, Qingdao University, Qingdao 266061, China
2   Institute of Finance and Economics, Shanghai University of Finance and Economics, Shanghai 200433, China
\*   Correspondence: sidkfinance@whu.edu.cn

**Abstract:** Appropriate social insurance contribution rates are crucial for the green development of firms. While the existing literature lacks an exploration of the relationship between social insurance policy and corporate environmental performance, this paper empirically examines the impact of social insurance contributions on corporate environmental performance using unbalanced panel data of 2947 A-share listed firms in China from 2008 to 2019. Our study shows that there is an inverted-U-shaped relationship between the social insurance burden and firms' environmental performance, and the result remains robust after changing the measurements of core variables, replacing estimation method, and controlling endogenous problems. The inverted-U-shaped relationship is more pronounced in non-heavily polluting industries, non-labor-intensive industries, and non-state-owned enterprises (non-SOEs). In addition, corporate innovation and digital transformation can positively moderate the inverted-U-shaped effect of social insurance burden on corporate environmental performance, and firms should grasp the "double-edged sword" effect of innovation and digital transformation in different periods of social insurance burden. Further analysis reveals that a reasonable social insurance burden can enhance firm value and risk taking through improving firms' environmental performance, whereas an excessive social insurance burden is not conducive to the improvement of firms' environmental performance, internal value creation, and risk taking.

**Keywords:** social insurance burden; environmental performance; innovation; digital transformation; enterprise value; risk taking

## 1. Introduction

As a "safety net" and "stabilizer" for people's livelihood, social security is crucial to stabilize employment, promote growth, and even achieve high-quality economic development. By the end of June 2021, China had established the world's largest social security system. Among the five main types of national social insurance (i.e., pension insurance, unemployment insurance, medical insurance, work injury insurance, and maternity insurance) in China, the basic pension insurance has covered 1.014 billion people, and the basic medical insurance has covered more than 1.3 billion people. As major participants in the modern market economy, firms have played a vital and active role in creating and perpetuating China's "economic growth miracle" over the past few decades, and have gradually become one of the main forces of China's social security. However, while firms are a mainstay of labor employment and social insurance contributions, they have long been regarded as the main source of environmental pollution. It has been a consensus that the resource-intensive development model of high energy consumption, high pollution, and high investment is no longer sustainable for China. In the face of China's increasingly prominent environmental problems and the increasingly severe environmental regulation situation, a growing number of firms have incorporated environmental performance into their long-term development strategies [1–3]. As the social insurance premiums paid by firms for their employees are calculated based on employees' total wages, firms have to

bear a higher burden of employment costs while providing social security benefits for their employees. In particular, in recent years, the proportion of firms' social insurance payment expenses to the labor cost of firms has reached a peak, far exceeding that of developed economies such as Europe and the United States [4]. The continuous burden of social insurance poses a potential risk to the normal operations and competitiveness of firms [5]. Under the constraints of China's current minimum wage system and the Social Insurance Law, the higher social insurance contribution rates and the ever-strengthening efforts made by governments to collect insurance premiums have put heavy financial pressure on Chinese firms, and the increasing burden of social insurance poses a major challenge to the high-quality development of Chinese firms. Therefore, how to achieve a better trade-off between social insurance contributions and the green development performance of firms has become a major practical issue to be faced by both governments and scholars.

At the micro level, early related studies mainly focused on the real effects of the social insurance burden on firms' labor force employment and wages [4,6], productivity [7], and cash holdings [8]. For example, Gruber and Krueger [6] and Nielsen and Smyth [4] found that, when firms pay more insurance premiums for their employees, they may incur higher costs, which will result in lower investment efficiency and even reduced employment. Zhang et al. [7] find that a high social insurance burden of firms will not only hinder technological progress, but also significantly reduce their total factor productivity, which is not conducive to the orderly operation and healthy development of enterprises. Deng et al. [8] found that rises in social insurance premiums in China promote labor-intensive firms to increase cash holdings.

Although existing studies have examined the micro effects of social insurance on the development of firms from different perspectives, they place more weight on short-term responses of firms to increases in social insurance contributions; there has been a failure to explore long-term strategic adjustments of the firms. In addition, the existing studies do not explore the potential channels through which social insurance burden affect firms' behaviors, such as innovation, risk taking, and investment efficiency. Moreover, limited attention has been paid to how to cope with the higher burden of social insurance and maximize the operational efficiency of firms. Given the increasing importance of environmental protection to firm value creation in recent years, rather limited work has considered the effect of social insurance burden on firms' environmental performance and its underlying mechanisms, which is not conducive to the high-quality development of real economy.

In view of this, this study is motivated to employ unbalanced panel data of Chinese A-share listed firms from 2008 to 2019 to assess the impact of social insurance contributions on corporate environmental performance. We not only identify the causal effect of social insurance contributions on corporate environmental performance, but also make a further exploration of the potential differences in effect of different types of firms, i.e., the heterogeneous effects of enterprises' social insurance burden on environmental performance, and the strategic responses of firms to increases in social insurance contributions from the perspective of corporate innovation and digital transformation. The empirical results show that there is an inverted-U-shaped relationship between firms' burden of social insurance contributions and their environmental performance. This conclusion remains valid after conducting several robustness tests, including replacing the core dependent variable or the independent variable, and using an instrumental variable approach. The results also show that both corporate innovation and digital transformation can positively moderate the inverted-U-shaped effect of social insurance contributions on firms' environmental performance. Moreover, the effect is more pronounced in non-heavily polluting industries, non-labor-intensive firms, and non-state-owned enterprises (Non-SOEs). Finally, we also found that a moderate and reasonable social insurance contribution rate not only helps to improve corporate environmental performance, but also promotes the growth of firm value through the enhancement of environmental performance. Our findings are helpful for governments to determine an appropriate social insurance contribution ratio and im-

prove effective social insurance payment methods, and thus to enhance the environmental performance of firms.

The main contributions of this study are threefold. First, this paper contributes to existing studies by exploring the effect of social insurance contributions on corporate environmental performance. Most studies have examined the effects of social insurance burden on macroeconomic variables, such as financial stability and economic development. However, the microeconomic effect of social insurance contributions has remained largely unstudied. In particular, existing studies lack have failed to investigate the relationship between social insurance contributions and corporate environmental performance. In this regard, this study fills the research gap by uncovering the real effects of social insurance contributions and the determinants of environmental performance of firms. On this basis, this study reveals an "inverted-U-shaped" relationship between the burden of social insurance contributions and the environmental performance of firms, which provides fresh insight for governments in China and other emerging economies in terms of dealing with the pressure of environmental protection. Second, this study provides additional evidence on how enterprise innovation and digital transformation affect the relationship between the burden of social insurance contribution and firm environmental performance. We find that firms' innovation and digital transformation could cope with the relationship of social insurance burden and firm corporate environmental performance. In other words, firms' innovation and digital transformation help to improve the positive impact of insurance burden on its environmental performance. Third, we verify the potential differences in the impact of social insurance contribution burden on environmental performance for firms with different characteristics. Therefore, policymakers should fully consider the differences among firms when formulating policies and measures regarding firms' social insurance contributions and environmental protection. To sum up, the findings of this study not only provide valuable real-world insights about how governments in China and other emerging economies improve social insurance system and promote green development of firms, but also provide lessons for firms in these economies to achieve a better trade-off between social security and green development.

The remainder of the study is structured as follows. Section 2 presents the theoretical analysis and research hypotheses. Section 3 presents the research design, including data sources, model specification, and variable definitions. Section 4 discusses the results of baseline regression and robust rests. Section 5 includes a heterogeneity analysis. Section 6 further examines the effect of corporate environmental performance and social insurance burden on firm value. Section 7 concludes.

## 2. Theoretical Analysis and Research Hypothesis

### 2.1. Corporate Social Insurance Contribution Burden and Environmental Performance

The social insurance system has become a critical vehicle for governments around the world to stabilize people's livelihood and enhance the level of national welfare [3]. However, social insurance contributions are important parts of firms' operating costs, and firms usually absorb this cost by passing it on to internal employees or external consumers. It is worth noting that, in a competitive market environment, however, most firms cannot control the external environment and do not have a monopoly position, so cannot fully pass on this cost to external consumers. At the same time, in China's current social security and minimum wage system, firms also cannot fully pass the cost on to internal employees [4]. From this point of view, social insurance contributions have become an unavoidable part of enterprises' costs.

Although participating in social insurance is a welfare provided by firms to their employees, it is essentially equivalent to payroll taxes and is an operating cost of firms, which not only affect their decisions of environmental behaviors to some extent, but also can incentivize firms to better control labor costs. On the one hand, paying social insurance for employees can generate an incentive effect. Environmental performance is an important manifestation of corporate environmental responsibility, which drives

firms to meet environmental legitimacy not only due to governmental supervision [9], but also to internal corporate responsibility. Social insurance contributions have quasi-tax characteristics, are required by governments, and will increase the labor costs of firms. With unavoidable social security expenditure and facing internal and external environmental constraints, firms devote themselves to improving labor productivity through a series of measures. In particular, firms will strive to improve their technological efficiency to achieve environmental compliance and sustainable development [10,11]. Second, increases in labor costs will prompt firms to actively reduce low-skilled labor and increase the training of high-skilled labor [12], and even increase capital investments to adopt more robots and other smart devices for digital transformation. It is worth noting that, as a result of strict requirements about pollution control and social insurance contributions, firms in heavily polluting industries mainly achieve indirect transfer of labor costs by optimizing human capital allocation and employee productivity, therefore providing a basis to stimulate their corporate environmental performance. In other words, under the pressure of social insurance contributions, firms will incentivize themselves to improve the structure of human capital, optimize the efficiency of human resource allocation and motivate firms to increase R&D expenditure and develop green technologies. Through the application of intelligent robots and IT technologies, the digital transformation of firms might be accelerated, which would gradually have a "substitution effect" on the demand for labor [13,14], thereby improving the environmental performance of firms.

On the other hand, the excessive burden of social insurance contributions magnifies the cost effect on firms. Compared with advanced economies, social insurance contribution rates in China remain at a higher level, which imposes a heavy financial burden on firms. As required by the China Labor Contract Law, firms must pay five types of insurance. Social insurance can provide a guarantee of the future livelihood of employees. However, a high social insurance contributions burden has a "crowding out" effect on employees' current disposable income [4], which would lead to lower employee motivation or corporate satisfaction and is not conducive to improving corporate environmental performance. Moreover, an excessive social security burden can also severely constrain corporate cash flows. This makes up an important cash flow expenditure, which can take funds away from production and investment. Although technological progress is the main driving force for productivity growth, technology development needs a large adjustment cost. Therefore, investments in technological innovation are more likely to be reduced, mainly because of the inherent risk and financing constraints of R&D. In other words, excessive payroll taxes will limit corporate investment in environmental management, R&D innovation, and digitalization [15–17], which prevents enterprises from engaging in improving new technologies, thus negatively affecting environmental performance.

In addition, the excessive burden of social insurance may also increase firms' tax avoidance behavior to avoid rising labor costs [18]. Despite alleviating the financial burden and cash flow constraints of firms to some extent, this may increase the risk of firms being punished for tax evasion, and further worsen the enthusiasm of firms for environmental protection. As social insurance payment is a compulsory and continuous cash flow expenditure, the cash outflows induced by social insurance will finally be derived from cash flows generated by a firm's operating activities. This will reduce the enterprises' free cash flows [19]. Though social insurance contributions could promote firms to optimize human resource allocation efficiency and labor productivity, it is hard to achieve noticeable progress in the short term. In contrast, if the social insurance contributions harm free cash flows, then private enterprises are prone to reduce investments in technology and productivity improvement. Based on the above analyses, we develop the first research hypothesis as follows:

**Hypothesis 1.** *There is an inverted-U-shaped nonlinear relationship between firms' burden of corporate social insurance and their environmental performance.*

## 2.2. The Moderating Role of Corporate Innovation

Corporate innovation is a core strategy for maintaining market competitiveness and creating sustainable value [18]. The change of labor cost may affect firms' investment decisions, particularly innovation decisions, by altering the cost of business activities. Moreover, the importance of innovation to the sustainable development of firms is self-evident, especially the far-reaching impact on the green transformation of the firms [19]. As mentioned above, the burden of social insurance affects the cash flows of firms, and innovation may be a "double-edged sword" to the environmental performance of firms due to the increase in the social insurance burden.

At the stage of lighter burden of social insurance contributions, firms, as the main contributors of economic development, fulfill their environmental responsibilities under the constraints of national environmental regulations. On the one hand, when implementing environmental protection strategies, firms need to increase corresponding inputs to research and develop new processes and technologies and enhance the intensity of technological innovation if they can freely cope with the burden of social insurance contributions [11]. On the other hand, firms paying social insurance premiums for their employees can be viewed as a means of investing in human capital rather than a mere cost pressure. As a kind of future security, social insurance can lead to human capital accumulation for firms, and is conducive to improving the green innovation ability of firms. By contrast, a higher level of social security can attract more outstanding high-skilled and high-quality talents for firms, thus optimizing their human capital structure and knowledge accumulation and ultimately promoting their innovation performance [20]. When the marginal environmental benefits from firms' innovation are larger than the marginal costs, firm innovation will reinforce the effect of the social insurance burden on environmental performance.

However, with the further aggravation of social insurance burden of firms, their labor costs may exceed the environmental performance benefits brought about by R&D and innovation investment. At the stage of high social insurance contributions, higher labor costs make firms face a heavy financial burden and cash flow constraints, which would in turn decrease profitability and increase financial constraints. Specifically, high labor costs make firms' operating leverage relatively high and make it more difficult for them to raise funds, thus forming severe financing constraints [21]. This may not only directly affect the investments in environmental management, but also have a "crowding-out effect" on R&D investment and technological innovation [16,22]. With tighter internal and external financial constraints, firms will have less ability to focus on environmental issues and might be likely to avoid taxes in order to neutralize the financial pressure of social insurance contributions [23]. In short, continuing to increase innovation and R&D will further increase firms' financial burden, which is not conducive to corporate environmental performance. Based on the above analysis, our second hypothesis is put forward as follows:

**Hypothesis 2.** *Corporate innovation positively moderates the relationship between firms' social insurance burden and environmental performance.*

## 2.3. The Moderating Role of Corporate Digital Transformation

Digitalization is playing an increasingly important role in improving the environmental performance of firms. In the era of digital economy, digital transformation has gradually become a common choice for firms to improve their competitiveness. In particular, the application of digital technologies, such as artificial intelligence, big data technology, and cloud computing, facilitates the collection, storage, transmission and identification of information affects the decision making of firms. Moreover, digital transformation has become a strategic imperative for firms to cope with fierce competition, and the impacts on firms' investment and financing decisions have attracted extensive attention [24]. Digitalization help firms effectively process massive information and enhance their efficiency in decision making. Firms experiencing digital transformation can use digital technologies to make

timely adjustments in response to shifts in economic conditions according to real-time information, thus avoiding unnecessary resources waste and improving the efficiency of production and operation. As a kind of payroll tax levied by governments, social insurance constitutes an important foundation of the national social security system. The relationship between the burden of social insurance and environmental performance of firms may be affected by the degree of digitalization transformation of firms when they cannot fully pass on the cost of social insurance.

When firms' burden of social insurance stays at a low level, rigid and irreversible social insurance contributions for employees will continue to raise the labor costs of firms, which will in turn increase the demand for digital technology and production aimed at replacing lower skilled workforce in order to improve the productivity. As labor costs increase and financial pressures rise in firms, advanced digital technologies and smart machines and equipment will be put into production in order to decrease the labor costs generated by social insurance [25]. In particular, when digital technology has more comparative advantages over the workforce, unskilled labor will be replaced by higher-level automated technologies, and such a substitution effect will lead to a decrease in the overall labor size and wages of firms [26,27]. It will enhance the sustainability of firms to convert labor into effective productivity, rather than directly reducing the number of employees to save the firms' capital. This also means that the digital transformation of firms can not only directly improve corporate environmental performance, but also indirectly increase resource inputs to improve environmental performance by converting sustained social security contributions into lower labor costs and more marginal profits.

However, the digital transformation of firms also induces continuous costs. When firms pay social insurance for their employees, it not only increases the incentives for employees, but also generates an operating cost. Moreover, digital transformation reduces firms' information asymmetry and increases their exposure to the market. Firms may attract more attention, which enhances their external pressures. Increases in pressure might result in more conservative managerial investment decisions, thereby promoting firms to invest in risky but profit-generating projects. However, innovation is a risky project and takes a long time; from this point of view, digital transformation may not be conducive to improving productivity in the short term. In addition, it is worth noting that, as the burden of social insurance contributions continues rising, the cost generated by the excessive burden of social insurance will seriously squeeze capital, which limits digital investment and inhibits the process of digital transformation. When the marginal benefits brought about by digital transformation in terms of the substitution of enterprise labor factors are not enough to cover the marginal labor costs caused by social insurance contributions, continuous inputs into digital transformation of firms will also further aggravate their financial constraints, which is detrimental to their environmental decisions and investments [28]. At the same time, the excessive burden of social security could make firms be busy with survival and neglect environmental business, which would in turn deteriorate their competitiveness and market value, further weakening corporate environmental performance [28]. Based on the above analyses, we present the third hypothesis as follows:

**Hypothesis 3.** *Digital transformation positively moderates the relationship between firms' burden of social insurance contributions and environmental performance.*

## 3. Research Design

### 3.1. Sample Selection

Since most Chinese firms started to disclose environmental information in 2008, the research sample of this study includes the Chinese firms listed on the Shanghai and Shenzhen stock exchanges in the period of 2008–2019. The annual financial data on these firms were obtained from the CSMAR database and RESSET database, and the city-level environmental regulation data were obtained from city government work reports.

To improve the validity of the sample and parameter estimation, we processed the original data in the following ways: (1) excluding firms with abnormal financial conditions such as ST and * ST; (2) excluding firms in the financial sector; (3) removing the firms with missing data in any of the main variables; (4) removing samples with a leverage ratio greater than 1, which are already insolvent; and (5) eliminating firms with less than three consecutive years in the sample. In addition, to avoid possible interference of extreme values with the estimation results, all firm-level continuous variables were Winsorized at the 1% and 99% levels. The final sample consisted of 21,325 unbalanced observations.

### 3.2. Model Specification

In order to empirically verify the impact of firms' social insurance contributions on their environmental performance suggested in Hypothesis 1, this study follows [29,30] to construct the following baseline econometric specification:

$$\text{Cep}_{i,t} = \alpha_0 + \alpha_1 \text{Insure}_{i,t-1} + \alpha_2 \text{Insure}^2_{i,t-1} + \alpha_3 X_{i,ct} + \gamma_i + \lambda_{jt} + \varepsilon_{i,jct} \tag{1}$$

where the subscripts i, c, t, and j indicate listed firms, cities, years, and industries, respectively; Cep indicates the environmental performance of a firm; and Insure is the social insurance contribution rate of the firm. X is the control variables, which mainly include firm size, employee size, firm age, director independence, return on assets, leverage ratio, corporate environmental management investment, and the level of environmental regulation in the city where the firm is located. $\gamma_i$ is the firm fixed effect, $\lambda_{jt}$ is the industry-year fixed effect, and $\varepsilon$ is the random disturbance term. $\alpha_1$ and $\alpha_2$ capture the nonlinear effect of the burden of corporate social insurance contributions on environmental performance.

Furthermore, social insurance contributions increase the labor cost of firms, and to a certain extent, they "force" firms to innovate or digitally transform in order to save labor cost, improve production processes, and achieve green production. In order to test the role of corporate innovation and digital transformation in the impact of social insurance contributions on environmental performance presented in Hypotheses 2 and 3, we followed [31] to include the interaction terms of corporate innovation, digital transformation, and social insurance contribution rate and its quadratic term in Equation (1) in the sense that:

$$\begin{aligned} \text{Cep}_{i,t} = \beta_0 + &\beta_1 \text{Insure}_{i,t-1} + \beta_2 \text{Insure}^2_{i,t-1} \\ &+ \beta_3 \text{Moder}_{i,t} + \beta_4 \text{Insure}_{i,t-1} \times \text{Moder}_{i,t} + \beta_5 \text{Insure}^2_{i,t-1} \times \text{Moder}_{i,t} + \beta_6 X_{i,ct} + \gamma_i + \lambda_{jt} \\ &+ \varepsilon_{i,jct} \end{aligned} \tag{2}$$

In Equation (2), Moder represents firm innovation and digital transformation in period t, and $\beta_4$ and $\beta_5$ reflect how firm innovation and digital transformation affect the relationship between the burden of social insurance and the environmental performance of firms, respectively.

### 3.3. Data and Descriptive Statistics

3.3.1. The Explained Variable

Among existing studies, there is no unanimity as to the measurement of corporate environmental performance; some studies use corporate pollution emissions, environmental management inputs, and emission costs to measure corporate environmental performance [29,32], but these measurements merely reflect some aspects of corporate environmental management or environmental protection. For Chinese firms, there is no authoritative organization to collect specific pollutant emission data. The environmental information disclosed by listed firms in three types of reports (specifically, annual reports, social responsibility reports, and environmental reports) is mostly textual and narrative, and only a few firms have disclosed specific pollutant emission data for many years, which cannot meet the conditions for empirical analysis in terms of sample size. Some recent studies employ "content analysis" to measure corporate environmental performance [9,30,33]. Therefore, based on previous research, this study constructs a comprehensive evaluation

index system to measure the environmental performance (Cep) of firms based on the "Guidelines on Environmental Information Disclosure of Listed Firms" released in 2010. The index system consists of 12 subitems, as displayed in Table 1, where each subitem is assigned a score of 1 (if Yes; 0 otherwise), and the total score of environmental performance of firms in a year ranges from 0 to 12.

**Table 1.** Enterprise environmental performance evaluation index system.

| No. | Attributes | Measurement |
|---|---|---|
| 1 | Whether the company's annual report is disclosed. | Yes = 1, No = 0 |
| 2 | Whether the corporate social responsibility report is disclosed. | Yes = 1, No = 0 |
| 3 | Whether the company's environmental report is disclosed. | Yes = 1, No = 0 |
| 4 | Whether the company's environmental protection concept is disclosed, such as disclosure of the firm's environmental protection concept, environmental policy, environmental management organization structure, circular economy development model, green development, etc. | Yes = 1, No = 0 |
| 5 | Whether the company's past and future environmental goals are disclosed. | Yes = 1, No = 0 |
| 6 | Has the company disclosed a series of management systems, systems, regulations, responsibilities, and other relevant environmental management systems? | Yes = 1, No = 0 |
| 7 | Whether the company participates in environmental education and training. | Yes = 1, No = 0 |
| 8 | Whether the company participates in environmental protection and other social welfare activities. | Yes = 1, No = 0 |
| 9 | Whether the company has established an emergency response mechanism for major environment-related emergencies, such as the emergency measures taken and the treatment of pollutants, etc. | Yes = 1, No = 0 |
| 10 | Whether the company has received environmental honors or awards. | Yes = 1, No = 0 |
| 11 | Whether the company has implemented the "Three Simultaneity" system, that is, the disclosure of the company's implementation of the "Three Simultaneity" system. | Yes = 1, No = 0 |
| 12 | Whether the company has passed ISO14001 certification. | Yes = 1, No = 0 |

3.3.2. The Explanatory Variable

Social insurance expenditure is part of the operating costs of firms, and the environmental behavioral decisions in the response of rising operating costs might lag. Therefore, we utilized the one-year lag of firms' social insurance contribution rate to measure the burden of social insurance (Insure). It is worth noting that using the one-year lag of social insurance contribution rate can also avoid the endogeneity problem caused by possible reverse causality between social insurance contributions and firm environmental performance at a certain degree.

Drawing on the research of Zhang et al. (2021) [7], we constructed the firm-level social insurance contribution rate of firms using the data of "employee compensation payable" and its line items in the notes to the financial statements of listed firms to represent the burden of social insurance contributions (Insure). Specifically, since the detailed accounts of "social insurance premiums" disclosed by firms are not uniform, and there are differences in the method of social insurance crediting, we extracted the data of "compensation payable to employees" and its detailed accounts from the notes to financial statements of listed firms from 2008 to 2019 from the RESSET database, and then manually compiled the data of "social insurance premiums" to ensure that all social insurance costs of the company for the year were covered and no other expenses were included. Finally, the ratio of the increase in "Salary Payable to Employees—Social Insurance" to the increase in "Salary Payable to Employees—Total" was used. A larger ratio of social insurance contribution suggests a greater burden of social insurance contributions faced by firms. China's Social Insurance Law clearly states that social insurance, which includes pension insurance, unemployment insurance, work injury insurance, medical insurance, and maternity insurance, is an impor-

tant social security system to promote social equity. At the same time, as an important part of the labor cost of firms, it also affects various behaviors of firms.

### 3.3.3. The Mechanism Variables

Following Hirshleifer et al. (2012) [34], we measured the level of firm innovation from the perspectives of R&D investment and patent output. We used the natural logarithm of 1 plus the total R&D inputs of firms to represent R&D investment (RD), and employed the natural logarithm of 1 plus the number of patents applied by firms to measure patent output (Patent). A larger value of Patent implies higher innovation ability.

Unlike existing studies that measure the digitalization based on the proportion of intangible assets [35], ICT investment [24], and whether or not digital transformation was conducted [5,36], this study used the natural logarithm of the sum of the frequency of digital transformation-related terms, including related keywords of "artificial intelligence technology," "block chain technology," "cloud computing technology," "big data technology," and "digital technology application" in the annual reports of firms to indicate the degree of digital transformation (Digital). These keywords involve many aspects of digital transformation of firms, which can reflect the level of their digital transformation to a certain extent. Furthermore, we adopted a combination of advanced machine learning method and text analysis to construct a firm-level digital transformation index, and employed it to measure firms' digital transformation level. Specifically, a higher frequency of a certain type of keywords in the annual report implies greater attention and more resources invested by firms in the digital arena.

### 3.3.4. The Control Variables

Drawing on related studies [7,9,11,30,32], we also controlled for other relevant variables that may affect corporate environmental performance. These variables include: (1) firm size (Size), represented by the natural logarithm of total assets of firms; (2) employee size (Labor), expressed as the natural logarithm of the number of employees of firms; (3) firm age (Age), measured as the natural logarithm of firm age; (4) the proportion of independent directors (Indep), measured by the ratio of the number of independent directors over the total number of directors; (5) return on assets (Roa), measured as the ratio of net profit over total assets; (6) leverage (Lev), represented as the ratio of total liabilities to assets; (7) environmental governance investment (Epinvest), represented by the natural logarithm of firms' environmental protection investment costs; and (8) city-level environmental regulation (ER), measured by the frequency of words related to environmental protection divided by the total number of words in the annual report on the work of the government of a city where the firm is located [37]. It is worth noting that environment-related terms include environmental protection, pollution, energy consumption, emission reduction, emissions, ecology, green, low carbon, air, chemical oxygen demand, sulfur dioxide, carbon dioxide, PM10, and PM2.5. In theory, larger firms are more susceptible to government scrutiny and public scrutiny and tend to actively engage in environmental initiatives [38,39], and firms with higher leverage face more severe financing constraints in terms of environmental governance and protection, which in turn would have a negative impact on their environmental performance [10]. Moreover, there are significant differences in the strategies adopted by old and young firms in terms of environmental protection [11]. With respect to internal governance, independent directors are able to perform a monitoring role in corporate decision making, thus influencing the environmental behavior of firms [40]. In addition, investments in environmental protection or pollution management can effectively promote the green and sustainable development of firms. The results of the descriptive statistics of the main variables are shown in Table 2.

**Table 2.** The description statistics.

| Variables | Mean | Sd. Dev. | Min. | P25 | P50 | P75 | Max. |
|---|---|---|---|---|---|---|---|
| Cep | 2.495 | 2.296 | 0.000 | 1.000 | 2.000 | 4.000 | 10.000 |
| Insure | 0.080 | 0.057 | 0.008 | 0.038 | 0.056 | 0.116 | 0.311 |
| RD | 17.815 | 1.459 | 12.206 | 16.952 | 17.805 | 18.670 | 22.170 |
| Patent | 3.756 | 2.174 | 0.000 | 2.485 | 4.111 | 5.257 | 8.829 |
| Digital | 3.115 | 1.364 | 0.000 | 2.079 | 2.996 | 4.007 | 6.547 |
| Size | 22.174 | 1.292 | 19.313 | 21.243 | 21.999 | 22.917 | 26.434 |
| Labor | 7.693 | 1.272 | 3.497 | 6.848 | 7.630 | 8.479 | 11.214 |
| Age | 2.745 | 0.393 | 0.693 | 2.565 | 2.833 | 2.996 | 3.526 |
| Indep | 0.374 | 0.054 | 0.250 | 0.333 | 0.333 | 0.429 | 0.600 |
| Roa | 0.039 | 0.058 | −0.431 | 0.015 | 0.036 | 0.065 | 0.226 |
| Lev | 0.432 | 0.206 | 0.035 | 0.267 | 0.427 | 0.590 | 0.896 |
| Epinvest | 0.000 | 0.002 | 0.000 | 0.000 | 0.000 | 0.000 | 0.029 |
| ER | 0.055 | 0.115 | 0.000 | 0.004 | 0.017 | 0.050 | 1.056 |

Note: This table provides descriptive statistics, namely, Mean, Std. dev. (standard deviation), Min. (minimum), P25 (25th percentiles), P50 (50th percentiles), P75 (75th percentiles), and Max. (maximum) for main variables used in our baseline regression and mechanism analysis.

## 4. Empirical Analysis

### 4.1. Baseline Regression

Table 3 reports the regression results of the relationship between the burden of corporate social insurance (Insure) and environmental performance (Cep). Column 1 shows the results without any control variables, and column 2 shows the results after adding the relevant control variables. We observe that the coefficient for the burden of enterprises' social insurance is significantly positive, and the coefficient of its quadratic term is significantly negative, indicating that there is an inverted-U-shaped relationship between firms' burden of social insurance and their environmental performance. The results of column 2 of Table 3 show that the coefficient for the burden of social insurance is 4.577, and its quadratic term is −11.124; we can therefore calculate that the turning point of the inverted-U shape is 0.206 ($-4.577/(-11.124 \times 2)$), which is located between the range of values of the burden of corporate social insurance contributions [0.008, 0.311]. This shows that there is a nonlinear relationship between the burden of corporate social insurance contributions and its environmental performance. The above results also mean that, when the social insurance contributions exceed 0.206, it will lead to a decline in environmental performance. In other words, there is a threshold in the relationship between enterprises' social insurance contributions and environmental performance. Combined with the mean value of firms' social insurance contributions of 0.080 in Table 2, we can further conclude that most of the firms in the sample still have large room to bear social security contributions and thus to improve their environmental performance. Thus, Hypothesis 1 is verified.

### 4.2. Endogeneity and Robustness Tests

#### 4.2.1. Endogeneity Tests

For the above regressions, we utilized the one-year lag of explanatory variables to test the effect of social insurance contributions on corporate environmental performance, which can overcome the potential reverse causality to a certain degree. However, there may still be endogeneity problems in the above empirical process. On the one hand, the potential omission of relevant control variables in the model may lead to a larger variance of the exogenous disturbance term and endogeneity problems. On the other hand, firms can decide their own social insurance contribution levels according to their own business ability and financial burden within the scope of China's social insurance policy; they could opt for self-selection in social insurance contributions, which may result in endogeneity problems to an extent. To this end, we mitigated for the potential endogeneity problem by the following methods.

**Table 3.** Baseline regression result.

| Variables | Cep | Cep |
|---|---|---|
| | (1) | (2) |
| Insure$_{t-1}$ | 10.324 *** | 4.577 *** |
| | (2.239) | (1.465) |
| Insure$^2_{t-1}$ | −20.207 *** | −11.124 *** |
| | (7.260) | (4.260) |
| Size | | 0.663 *** |
| | | (0.040) |
| Labor | | 0.183 *** |
| | | (0.036) |
| Age | | 0.184 ** |
| | | (0.083) |
| Indep | | −0.726 |
| | | (0.474) |
| Roa | | 0.730 ** |
| | | (0.363) |
| Lev | | −0.566 *** |
| | | (0.169) |
| Epinvest | | 127.404 *** |
| | | (11.379) |
| ER | | 0.688 ** |
| | | (0.271) |
| Constant | 1.866 *** | −13.976 *** |
| | (0.117) | (0.752) |
| Firm fixed effects | Yes | Yes |
| Industry fixed effects | Yes | Yes |
| R$^2$ | 0.108 | 0.293 |
| N | 21,324 | 21,324 |

Note: ** and *** indicate significance at 5% and 1% significance levels, respectively. The numbers in the parenthesis are robust standard errors.

In order to test the robustness of our main conclusion, we followed [41,42] to employ the instrumental variable method, and utilized the one-year lag of the mean value of social insurance contribution rates of other firms in the same industry and in the same region as the instrumental variable of the burden of social insurance (Insure). The instrumental variable satisfies the conditions of relevance and exogeneity. Specifically, under the guidance of China's social insurance policy, each province has a certain degree of autonomy in formulating specific social insurance implementation plans. Therefore, firms in the same province face the same policy regulations and reference range, and the correlation is high within the same city, even though different firms have different actual social insurance contribution rates. Thus, the instrumental variable satisfies the assumption of correlation with the explanatory variables. Meanwhile, the mean value of social insurance contribution rates of other firms in the same industry and in the same region does not affect a firm's corporate environmental performance directly; therefore, the instrumental variable satisfies the exclusiveness and exogeneity. The results in column 1 of Table 4 display that the results of the IV-GMM test based on the instrumental variable are similar to the results of the baseline regression, i.e., there is an inverted-U-shaped effect of the burden of corporate social insurance contributions on environmental performance. Meanwhile, the *F* value of the unidentifiable test in the first stage of the instrumental variable is 89.473, corresponding to a p-value of 0.000, which leads us to strongly reject the null hypothesis of unidentifiability. The Hansen J statistic used to test for overidentification has a value of 0, thus there is no overidentification, which proves that the model is aptly identified. In addition, the Kleibergen–Paap rk Wald *F* value for the weak instrumental variable test is 44.487, which corresponds to a critical value of 7.03 for the Stock–Yogo weak instrumental variable test at the 10% level, leading us to reject the hypothesis that the instrumental variable is weak. All of the above tests support the validity of the instrumental variables selected in this study.

**Table 4.** Results of robustness tests.

| Variables | Cep | Cep | Cep | Cep | Cep | Cep | Cep | Cep |
|---|---|---|---|---|---|---|---|---|
| | (1) | (2) | (3) | (4) | (5) | (6) | (7) | (8) |
| $Insure_{t-1}$ | 14.442 *** | 8.471 *** | 1.045 | 1.383 *** | 4.648 *** | 4.919 *** | 2.594 ** | 4.902 ** |
| | (3.462) | (1.908) | (1.100) | (0.530) | (1.406) | (1.623) | (1.178) | (2.310) |
| $Insure_{t-1}^2$ | −49.305 *** | −20.295 *** | −3.796 | −0.458 ** | −10.959 *** | −12.359 ** | −5.889 * | −14.576 *** |
| | (12.628) | (5.479) | (2.618) | (0.220) | (3.935) | (4.858) | (3.037) | (5.049) |
| Size | 0.627 *** | 0.717 *** | 0.671 *** | 0.684 *** | 0.577 *** | 0.633 *** | 0.711 *** | 0.709 *** |
| | (0.025) | (0.052) | (0.045) | (0.044) | (0.046) | (0.040) | (0.048) | (0.070) |
| Labor | 0.202 *** | 0.177 *** | 0.189 *** | 0.188 *** | 0.272 *** | 0.169 *** | 0.127 *** | 0.149 ** |
| | (0.022) | (0.049) | (0.040) | (0.039) | (0.043) | (0.036) | (0.041) | (0.067) |
| Age | 0.252 *** | 0.098 | 0.133 | 0.155 | 0.167 ** | 0.210 ** | 0.067 | −0.411 * |
| | (0.048) | (0.119) | (0.094) | (0.096) | (0.084) | (0.089) | (0.085) | (0.215) |
| Indep | −0.653 ** | −0.000 | −0.673 | −0.790 | −0.676 | −0.614 | −0.716 | 0.037 |
| | (0.310) | (0.688) | (0.541) | (0.521) | (0.474) | (0.493) | (0.584) | (0.974) |
| Roa | 1.239 *** | 0.636 | 0.849 * | 0.620 | 0.528 | 0.852 ** | 0.169 | −0.587 |
| | (0.285) | (0.518) | (0.446) | (0.391) | (0.368) | (0.369) | (0.670) | (0.920) |
| Lev | −0.523 *** | −0.747 *** | −0.674 *** | −0.738 *** | −0.546 *** | −0.516 *** | −0.726 *** | −1.486 *** |
| | (0.105) | (0.233) | (0.189) | (0.189) | (0.169) | (0.179) | (0.202) | (0.351) |
| Epinvest | 122.247 *** | 117.321 *** | 140.270 *** | 131.055 *** | 126.234 *** | 120.453 *** | 552.882 *** | 130.201 *** |
| | (11.977) | (20.121) | (13.795) | (12.353) | (11.497) | (11.119) | (58.584) | (17.902) |
| ER | 0.066 | 0.245 | 0.647 ** | 0.686 ** | 0.797 *** | 0.443 | 0.682 ** | 0.923 * |
| | (0.248) | (0.414) | (0.290) | (0.297) | (0.274) | (0.274) | (0.298) | (0.528) |
| Insure_Policy | | | 9.298 *** | | | | | |
| | | | (2.392) | | | | | |
| Insuresq_Policy | | | −28.395 *** | | | | | |
| | | | (8.147) | | | | | |
| Pwage | | | | | 0.241 *** | | | |
| | | | | | (0.062) | | | |
| Constant | −14.889 *** | −16.640 *** | −15.328 *** | −14.951 *** | −16.239 *** | −13.810 *** | −15.488 *** | −13.233 *** |
| | (0.446) | (1.049) | (0.933) | (0.878) | (0.894) | (0.793) | (0.958) | (1.512) |
| Firm fixed effects | Yes | Yes | Yes | Yes | Yes | Yes | Yes | Yes |
| Industry fixed effects | Yes | Yes | Yes | Yes | Yes | Yes | Yes | Yes |
| Year-Industry fixed effects | | | | | Yes | | | |
| $R^2$ | 0.297 | 0.352 | 0.301 | 0.283 | 0.301 | 0.258 | 0.273 | 0.319 |
| N | 12,585 | 9747 | 18,030 | 17,861 | 21,324 | 17,886 | 9540 | 7425 |

Note: *, **, and *** indicate significance at 10%, 5%, and 1% significance levels, respectively. The numbers in the parenthesis are robust standard errors.

In order to maintain the robustness of the above results, we further investigated firms located in regions where social insurance contributions are fully collected by taxation departments as an alternative sample. At present, China's enterprise social insurance fees form a dichotomous coexistence pattern of taxation and social insurance agency collection. In 2018, for example, provinces where enterprise social insurance is fully collected by the taxation department accounted for more than two-thirds of those in China, and in less than one-third of the other provinces it was collected by social insurance agencies. Compared with the social insurance department, the taxation department is more familiar with financial information such as enterprise wages and profits, and has the advantage of authority in policy implementation, which is conducive to strengthening social insurance collection, protecting workers' rights and interests, and preventing firms' tax evasion and omission of payment. Therefore, when the taxation department is used as the enterprise social insurance collection department, the enterprise social insurance contribution rate is basically not affected by its own financial situation, which effectively enhances the exogenous nature of the enterprise social insurance fee payment. For this reason, firms in regions where social insurance is fully collected by the taxation department are further

selected as the sample for robustness testing, and the results in column 2 of Table 4 are consistent with the results of the basic test, indicating that the conclusion of the inverted-U-shaped effect of the burden of enterprise social insurance contributions on environmental performance holds.

Furthermore, the Chinese government issued the Social Insurance Law in 2011, which improved the collection and control of corporate social insurance contributions and effectively addressed the widespread phenomenon of social insurance evasion and underpayment. Considering that the enactment and implementation of this law is not subject to individual firm behaviors and is a relatively exogenous shock event for corporate social insurance contributions, we included the Social Insurance Law (Policy) and its interaction terms with the primary and secondary terms of social insurance contributions (Insure_Policy, Insuresq_Policy) in Equation (1), and excluded the firms listed after 2011, in order to test whether the implementation of this policy strengthens the impact of corporate social insurance contribution burden on environmental performance. When employing the sample after 2011, we set Policy to 1; otherwise, Policy = 0. The results in column 3 of Table 4 present that the coefficients of Insure_Policy and Insuresq_Policy are 9.298 and −28.395, respectively, which both pass the significance test at the 1% level, indicating that the implementation of the Social Insurance Law strengthened the nonlinear impact of firms' social insurance contribution burden on environmental performance.

### 4.2.2. Robustness Tests

(1) Changing the measurement of explanatory variables

Since the level of social insurance contributions is determined based on the average wages of firms, the ratio of "Salary Payable to Employees—Social Insurance" to "Salary Payable to Employees—Total" reflects the ratio of social insurance contributions to total compensation expenses. Therefore, the employee training and welfare expenses included in "Salary Payable to Employees" do not constitute the base for calculating social insurance contributions, which may lead to bias in the explanatory variable measurement and further influence the robustness of empirical results. In addition, the ratio of the increase in "Salary Payable to Employees—Social Insurance" to the increase in "Salary Payable to Employees—Wages, Bonuses, Allowances, and Subsidies" is used to measure the burden of social insurance contributions. The results in column 4 of Table 4 show that the inverted-U-shaped relationship between the burden of social insurance contributions and the environmental performance of firms still holds after changing the measurement of the explanatory variables.

(2) Changing the measurement of dependent variable

In the above baseline analysis, we adopted a text analysis to construct the environmental performance index and took it as the core dependent variable. In this section, we adopted the ratio of environmental expenditure to total cost as the alternative measure of dependent variable. The results in column 5 of Table 4 display that the coefficient of enterprises' social insurance contributions on environmental performance is still significantly positive, and the coefficient of quarter term of social insurance contribution is significantly negative, which suggests the robustness of the above results.

(3) Changing the model setting

On the one hand, considering that the differences of some industry characteristics may be dynamically adjusted to the change of time trend, the regression further controls for the "Year × Industry" fixed effect to control for the unobservable factors of time variation in the industry level of firms—for example, some industry policies may affect the baseline empirical results; therefore, we further controlled the industry-year effect to clearly identify the effect of social insurance burden on corporate environmental performance. On the other hand, the social insurance contributions of firms have a strong relationship with the wage level, so we further controlled for the per capita wage and salary of firms in the regression to exclude the potential influence of wage level changes on the findings of this study. In other words, we added further control variables to avoid omitting key variables to keep

the results robust. The results in column 5 of Table 4 show that, after controlling for the "Year × Industry" effect and enterprise wage level, the nonlinear effect of enterprise social insurance contribution burden on environmental performance was in line with the baseline results, which supports the robustness of the basic findings.

(4) Excluding other shocks and changing the sample

It is worth noting that not all firms disclose relevant environmental information. In order to improve the validity of the empirical results, we further excluded those samples of firms with zero environmental performance for robustness testing; the estimation results are displayed in column 6 of Table 4. Moreover, the new Environmental Protection Law of China in 2015 included the ecological protection red line for the first time, explicitly requiring key emission units or firms to truthfully disclose their main pollutant names, emission methods, and concentration to the public, which may have had an impact on the conclusions of this study. Therefore, we also excluded the samples from 2015 and later years from the robustness test, and the estimation results are shown in column 7 of Table 4. Furthermore, in order to enhance the validity and comparability of the findings, the unbalanced panel was further transformed into a balanced panel to ensure that the observations of each firm were in the same dimension; the estimation results are shown in column 8 of Table 4. We observe that the above results did not fundamentally change, which proves the robustness of the conclusion that there is an inverted-U-shaped effect of enterprise social insurance contribution burden on environmental performance.

*4.3. Mechanism Analysis*

When the effect we examined in Equation (1) presented a nonlinear relationship, for Equation (2) with the introduction of the mechanism variable (Moder), we mainly observed whether the coefficient $\beta_5$ of the interaction term between the quadratic term of the explanatory variable and Moder was significant [31]. When the main effect is an inverted-U-shaped relationship, if $\beta_5 < 0$, it means that the mechanism variable enhances the nonlinear effect of the main relationship of first promoting and then suppressing, which is manifested as the curve arc on both sides of the inflection point becomes steeper; on the contrary, if $\beta_5 > 0$, it means that the mechanism variable weakens the nonlinear effect of first promoting and then inhibiting the main relationship, when the curve arc on both sides of the inflection point becomes flatter. As mentioned in the previous section, the burden of corporate social insurance contributions can influence the environmental performance of firms through corporate innovation and digital transformation. The estimation results based on Equation (2) to validate this logic are presented in Table 5.

The results in column 1 of Table 5 show that the coefficient of the interaction term between R&D investment (RD) and social insurance contribution burden is significantly positive, whereas the coefficient for the interaction term with the quadratic term of social insurance contribution burden is significantly negative, suggesting that R&D investment reinforces the inverted-U-shaped effect of the social insurance contribution burden on the environmental performance of firms. Similarly, the results in column 2 of Table 5 show that the coefficient of the interaction term between patent output (Patent) and social insurance contribution burden is significantly positive, while the coefficient of the interaction term of patent output and the secondary term of social insurance contribution burden is significantly negative, implying that corporate innovation ability also strengthens the nonlinear effect of the social insurance contribution burden on environmental performance. The above facts indicate that the increase in the corporate social insurance contribution rate can urge these firms to strengthen research and development, improving the level of innovation ability, so as to further improve environmental performance. During the period of the rising burden of corporate social insurance contributions, the environmental benefits brought by firms with higher levels of innovation can cover the loss of environmental input costs due to social insurance expenditures, and create incentives for environmental performance improvement through innovative methods such as increased R&D investment and higher innovation patent output. However, with the further increase in the burden of social

insurance contributions, the high cost of labor protection for firms has a "squeezing effect" on innovation inputs. The negative impact of the burden of social insurance contributions on environmental performance at this stage is exacerbated by the fact that firms' innovation inputs are financially constrained, their innovation outputs are hampered, and the marginal benefits generated by their lack of innovation capacity cannot cover the negative marginal costs caused by social insurance expenditures. Hypothesis H2 is thus verified.

**Table 5.** Results of moderation effects.

| Variables | Cep | Cep | Cep |
|---|---|---|---|
| | (1) | (2) | (3) |
| $Insure_{t-1}$ | −18.669 | 0.375 | −3.776 ** |
| | (16.171) | (2.982) | (1.903) |
| $Insure_{t-1}^2$ | 111.182 * | 11.005 | 12.903 ** |
| | (65.100) | (10.959) | (6.258) |
| RD | −0.035 | | |
| | (0.051) | | |
| $Insure_{t-1} \times RD$ | 1.699 * | | |
| | (0.949) | | |
| $Insure_{t-1}^2 \times RD$ | −8.286 ** | | |
| | (3.881) | | |
| Patent | | −0.110 *** | |
| | | (0.033) | |
| $Insure_{t-1} \times Patent$ | | 2.488 *** | |
| | | (0.753) | |
| $Insure_{t-1}^2 \times Patent$ | | −10.965 *** | |
| | | (3.455) | |
| Digital | | | −0.220 *** |
| | | | (0.035) |
| $Insure_{t-1} \times Digital$ | | | 2.986 *** |
| | | | (0.610) |
| $Insure_{t-1}^2 \times Digital$ | | | −9.040 *** |
| | | | (2.743) |
| Size | 0.599 *** | 0.640 *** | 0.656 *** |
| | (0.052) | (0.062) | (0.040) |
| Labor | 0.252 *** | 0.307 *** | 0.202 *** |
| | (0.048) | (0.062) | (0.036) |
| Age | 0.349 *** | 0.364 *** | 0.161 * |
| | (0.099) | (0.115) | (0.084) |
| Indep | −0.886 | −1.262 ** | −0.731 |
| | (0.539) | (0.635) | (0.472) |
| Roa | 0.803 ** | 1.498 *** | 0.682 * |
| | (0.399) | (0.511) | (0.363) |
| Lev | −0.460 ** | −0.318 | −0.570 *** |
| | (0.208) | (0.256) | (0.170) |
| Epinvest | 129.449 *** | 131.157 *** | 122.914 *** |
| | (11.768) | (13.801) | (11.396) |
| ER | 0.752 ** | 0.413 | 0.610 ** |
| | (0.307) | (0.367) | (0.273) |
| Constant | −12.974 *** | −14.286 *** | −13.206 *** |
| | (1.107) | (1.081) | (0.761) |
| Firm fixed effects | Yes | Yes | Yes |
| Year-Industry fixed effects | Yes | Yes | Yes |
| $R^2$ | 0.297 | 0.317 | 0.295 |
| N | 15,353 | 9645 | 21,168 |

Note: *, **, and *** indicate significance at 10%, 5%, and 1% significance levels, respectively. The numbers in the parenthesis are robust standard errors.

Column 3 of Table 5 shows the results that include the moderating effect of enterprise digital transformation. The coefficient of $\text{Insure}_{t-1} \times \text{Digital}$ is 2.986 and the coefficient of $\text{Insure}_{t-1}^2 \times \text{Digital}$ is $-9.040$, both of which are significant at the 1% level, indicating that digital transformation has a reinforcing effect on the inverted-U-shaped relationship between the burden of corporate social insurance contributions and environmental performance. With the rising proportion of social insurance contributions of firms, firms seek digital operation or management to replace the low-skilled labor force of repetitive work in order to make up for the loss of labor costs. Firms with a higher level of digitization can not only reduce social insurance expenses, but also enable firms to invest more energy and resources to improve environmental performance, resulting in a strengthening effect. However, when the enterprise social insurance contribution rate exceeds the threshold value of the inverted-U shaped curve, and when the burden of excessive and rigid social insurance contributions cannot be fully shifted due to the existence of national social insurance coercive force, it will not only make the enterprise digital transformation face challenges, but also make the enterprise environmental performance improvement constrained. From this point of view, the digital transformation of firms can exacerbate the negative environmental effects of the burden of social insurance contributions on firms. The above empirical facts suggest that digital transformation positively moderates the inverted-U-shaped effect of corporate social insurance contribution burden on environmental performance. Based on the above empirical results and analysis, Hypothesis H3 is verified.

## 5. Heterogeneity Analysis

In the above analysis, we did not consider the potential differences in the effect of social insurance contribution burden on environmental performance among firms. However, as mentioned above, for different regions and industries, the institutional and operational characteristics are different, such as production factor patterns, property rights properties, and so on, which may also lead to potential differences in the effect of social insurance burden on corporate environmental performance. Therefore, we next performed a heterogeneity analysis, aiming for a detailed understanding of the impact of the social insurance contribution burden on firms' environmental performance.

### 5.1. The Role of Different Industries

Heavily polluting industries face more environmental constraints, higher external monitoring, and higher financial constraints than non-heavily polluting industries, which may lead to differences in the impact of social insurance contribution on corporate environmental performance. In particular, at low environmental performance levels, firms in heavily polluting industries are more willing to proactively implement environmental governance policies to achieve environmental compliance than firms in non-heavily polluting industries [43]. In fact, as firms in heavily polluting industries face higher environmental management costs and stress risks, they will adopt more proactive environmental protection strategies under strict environmental controls, and with the improvement in the level of environmental performance, these strategies make it easier for firms to receive subsidies from the local government. China's environmental protection law supports rewards for units or individuals that achieve significant environmental protection. At the same time, the environmental compensation received by firms will weaken the negative impact of the rising burden of social insurance contributions on environmental performance to an extent. Therefore, the positive and negative effects of the social contribution burden on the environmental performance of firms in heavily polluting industries are relatively insignificant compared with those in non-heavily polluting industries. Based on the list of firms in heavily polluting industries issued by the relevant Chinese government departments, we divided the whole sample into firms in heavily polluting industries (Hpi = 1) and non-heavily polluting industries (Hpi = 0). The results in columns 1 and 2 of Table 6 show that the coefficients for the burden of corporate social insurance contributions and its quadratic term are more significant among firms in non-heavily polluting industries at the 1% level.

Since the coefficients in different groups cannot be directly compared, the Chow test was employed to examine the difference in coefficients between groups. We found that there was a significant difference in the nonlinear effect of social insurance contribution burden on environmental performance between the two groups. These results suggest that the inverted-U-shaped relationship of social insurance contribution burden and environmental performance was more manifest for firms located in non-heavily polluting industries than those in heavily polluting industries.

**Table 6.** Heterogeneity analysis.

| Variables | Hpi = 1 | Hpi = 0 | Highlabor = 1 | Highlabor = 0 | Soe = 1 | Soe = 0 |
|---|---|---|---|---|---|---|
| | (1) | (2) | (3) | (4) | (5) | (6) |
| $Insure_{t-1}$ | 1.891 | 4.906 *** | 3.006 ** | 8.127 *** | −2.570 | 6.650 *** |
| | (1.930) | (1.654) | (1.525) | (2.358) | (1.977) | (1.423) |
| $Insure_{t-1}^2$ | −4.639 * | −16.372 *** | −8.032 ** | −21.884 *** | −0.234 | −16.476 *** |
| | (2.702) | (4.967) | (3.848) | (7.069) | (2.122) | (3.817) |
| Size | 0.612 *** | 0.519 *** | 0.653 *** | 0.650 *** | 0.787 *** | 0.631 *** |
| | (0.073) | (0.040) | (0.054) | (0.055) | (0.087) | (0.041) |
| Labor | 0.343 *** | 0.278 *** | 0.146 *** | 0.229 *** | 0.134 | 0.198 *** |
| | (0.074) | (0.037) | (0.048) | (0.050) | (0.082) | (0.036) |
| Age | −0.057 | 0.063 | 0.231 ** | 0.127 | −0.364 ** | 0.257 *** |
| | (0.184) | (0.086) | (0.098) | (0.119) | (0.179) | (0.086) |
| Indep | −1.470 | −0.058 | −1.068 * | −0.484 | −1.818 * | −0.561 |
| | (0.908) | (0.559) | (0.599) | (0.645) | (1.055) | (0.494) |
| Roa | 0.213 | 0.647 * | 0.764 * | 0.564 | −0.444 | 0.894 ** |
| | (0.840) | (0.370) | (0.404) | (0.598) | (1.256) | (0.366) |
| Lev | −0.724 ** | −0.577 *** | −0.107 | −1.058 *** | −1.127 *** | −0.477 *** |
| | (0.347) | (0.173) | (0.213) | (0.241) | (0.430) | (0.170) |
| Epinvest | 104.603 *** | 165.906 *** | 147.805 *** | 112.018 *** | 121.323 *** | 127.385 *** |
| | (12.746) | (22.088) | (19.130) | (13.341) | (23.166) | (12.407) |
| ER | 0.639 * | 0.305 | 0.513 | 0.946 ** | 0.631 | 0.719 *** |
| | (0.351) | (0.447) | (0.343) | (0.369) | (0.632) | (0.265) |
| Constant | −12.387 *** | −11.626 *** | −13.529 *** | −13.910 *** | −13.709 *** | −13.761 *** |
| | (1.375) | (0.752) | (1.014) | (1.054) | (1.658) | (0.780) |
| Chow Test | 74.64 *** | | 12.99 *** | | 11.02 *** | |
| Firm fixed effects | Yes | Yes | Yes | Yes | Yes | Yes |
| Year-Industry fixed effects | | Yes | Yes | Yes | Yes | Yes |
| $R^2$ | 0.262 | 0.244 | 0.260 | 0.312 | 0.328 | 0.289 |
| N | 7258 | 14,067 | 10,658 | 10,666 | 3058 | 18,266 |

Note: *, **, and *** indicate significance at 10%, 5%, and 1% significance levels, respectively. The numbers in the parenthesis are robust standard errors.

### 5.2. The Role of Production Factor Patterns

Enterprises' social insurance contributions expenditure is a main part of labor cost. In theory, labor-intensive firms are highly dependent on labor, and their response to labor cost increases is relatively sensitive. Moreover, labor-intensive firms are more likely to make decisions to increase or decrease their number of employees based on changes in labor costs. From this point of view, the effect of the social insurance contribution burden on the environmental performance of labor-intensive firms is expected to be limited relative to that of non-labor-intensive firms. Drawing on Serfling (2016) [44], the ratio of total cash paid to and for employees to the firm's operating income is used to measure the firm's labor intensity. When firms' labor intensity is greater than the median labor intensity of all firms in that year, the firms are classified as labor-intensive (Highlabor = 1); otherwise, they are classified as non-labor-intensive (Highlabor = 0). The results in columns 3 and 4 of Table 6 show that the coefficients of social insurance contribution burden and its quadratic term pass the 1% significant level in the sample of non-labor-intensive firms, and the effect of a rising social insurance contribution ratio on the environmental performance of non-labor-intensive

firms in terms of first promoting and then inhibiting is relatively more significant. In addition, the results of the intergroup coefficient difference test (i.e., Chow test) show that the impact of the social insurance contribution burden on the environmental performance of firms is significantly different between the two groups. The level of social insurance contributions of Chinese firms remains at a relatively high level at the present stage. Meanwhile, under the increasing compulsion and regulation of national social insurance contributions, labor-intensive firms will hold more liquidity in the face of changes in social insurance contribution burden [8], which would hinder the enhancement of environmental performance. In contrast, non-labor-intensive firms, which typically have higher human capital accumulation, will be encouraged to actively engage in R&D investment and digital transformation, to compensate for the "squeeze" on environmental costs from rising labor costs. However, as labor costs rise disproportionately, an increase in social insurance contribution rates can also significantly inhibit the environmental performance of non-labor-intensive firms.

### 5.3. The Role of Property Rights

For China's economic and social development, state-owned enterprises (SOEs) take more responsibility for solving employment and environmental protection compared with non-SOEs [45]. Moreover, SOEs have advantages in terms of accessing government policy support and bank credit, which results in differences in mitigating enterprise operating costs. From this point of view, the nonlinear relationship between the burden of corporate social insurance and environmental performance may differ between the two kinds of firms. In order to verify the difference in this effect between firms with different types of property rights, we defined the firms as SOEs (Soe = 1) when their actual controller is the state-owned sector or state-owned legal person, and as non-SOEs (Soe = 0), otherwise. The results in columns 5 and 6 of Table 6 display that the coefficients for the burden of enterprise social insurance and its quadratic term are not significant in SOEs, while the coefficients of social insurance contribution and its quadratic term in the non-SOEs subsample are 6.650 and −16.476, respectively, which are significant at the 1% level. The results show that non-SOEs' social insurance burden has a more significant inverted-U-shaped effect on their environmental performance. Meanwhile, the results of the Chow test indicate that there is a significant difference in the nonlinear effect of social insurance contribution burden on the environmental performance of firms between these two subsamples. The results are closely related to the fact of national development that SOEs have strong state credit backing and receive relatively more state financial investment and policy support [46]. At the same time, SOEs also take more responsibility for compliance and are subject to more supervision from the government, the public, and the media [9]. At low levels of environmental performance, SOEs are more likely to increase their environmental monitoring and compliance enforcement to gain environmental legitimacy than private firms; this enforcement tends to be mandatory rather than incentive-based, and managers of SOE are less motivated to pursue environmental performance. When firms are at a higher level of environmental performance, SOEs allocate more resources to relatively risky projects than non-SOEs, and SOEs receive state financial compensation or policy support for risk, which weakens the "crowding-out" effect of social insurance contributions on environmental investment. That is, the negative relationship of the burden of social insurance contributions and environmental performance at higher levels of environmental performance is weakened. Therefore, compared with state-owned firms, the impact of social insurance contribution burden on environmental performance will be more significant in non-state-owned firms.

## 6. Further Analysis

Good environmental performance is an important driving force of enterprises' value growth. Our previous analyses have confirmed that there is a nonlinear effect of the burden of corporate social insurance contributions on environmental performance. So, can this



effect be transmitted to the level of corporate value and affect corporate value growth and risk taking? To test this question, we attempted to further explore the impact of improved environmental performance on the future value of the firms.

Enterprise value is not only an important investment orientation for firms to obtain market resources, but also a long-term strategic goal of enterprise management [47]. We used Tobin's Q of firms to measure firm value, which is expressed as the ratio of the market value of the firms to the replacement cost of capital. For firms' risk-taking, we utilized a three-period rolling standard deviation of ROA. Following existing studies [2,15], we constructed interaction terms between the burden of corporate social insurance and its quadratic term and environmental performance (Cep) and then regressed them on firm value. The results in Table 7 show that the coefficients of $\text{Insure}_{t-1} \times \text{Cep}$ were significantly positive, and the coefficients of $\text{Insure}^2_{t-1} \times \text{Cep}$ were significantly negative in the regressions of firm value and risk taking. This indicates that a moderate corporate social insurance contribution burden can promote the growth of firm value and risk taking through improved environmental performance, but an excessive social insurance contribution burden can deteriorate the growth of corporate value and risk taking through reducing environmental performance. This further suggests that setting a reasonable and moderate corporate social insurance contribution ratio is not only helpful to improve corporate environmental performance, but also to further enhance firm value and risk taking through improving environmental performance.

**Table 7.** Corporate social insurance contribution burden, environmental performance and corporate value.

| Variables | Tobin's Q | Risk-Taking |
|---|---|---|
| | (1) | (2) |
| $\text{Insure}_{t-1} \times \text{Cep}$ | 0.972 *** | 0.033 *** |
| | (0.357) | (0.001) |
| $\text{Insure}^2_{t-1} \times \text{Cep}$ | −3.367 ** | −1.224 *** |
| | (1.564) | (0.016) |
| $\text{Insure}_{t-1}$ | −1.132 | 0.144 |
| | (1.221) | (1.010) |
| $\text{Insure}^2_{t-1}$ | 7.664 * | 3.551 *** |
| | (4.656) | (0.151) |
| Cep | −0.021 | 0.133 |
| | (0.015) | (1.028) |
| Size | −0.454 *** | −0.155 *** |
| | (0.025) | (0.000) |
| Labor | −0.013 | 0.027 |
| | (0.019) | (0.029) |
| Age | 0.248 *** | 0.163 *** |
| | (0.042) | (0.012) |
| Indep | 1.118 *** | 1.042 *** |
| | (0.259) | (0.020) |
| Roa | 3.756 *** | 2.656 *** |
| | (0.343) | (0.035) |
| Lev | −0.051 | 0.016 |
| | (0.114) | (0.027) |
| Epinvest | −10.196 *** | −1.052 *** |
| | (2.798) | (0.001) |
| ER | −0.238 * | 0.026 ** |
| | (0.125) | (0.000) |
| Constant | 10.986 *** | 2.105 *** |
| | (0.471) | (0.015) |
| Firm fixed effects | Yes | Yes |
| Year-Industry fixed effects | Yes | Yes |
| $R^2$ | 0.354 | 0.522 |
| N | 21,324 | 21,186 |

Note: *, **, and *** indicate significance at 10%, 5%, and 1% significance levels, respectively. The numbers in the parenthesis are robust standard errors.

## 7. Conclusions

Green development is crucial to corporate value growth, and exploring the impact of corporate social insurance contribution rates on environmental performance has positive implications for improving the current social insurance system. Using annual financial data on A-share listed firms in China during the period of 2008–2019, we empirically identified the impact of corporate social insurance contribution burden on environmental performance and its mechanism. The results show that there is an inverted-U-shaped effect of firms' social insurance burden on their environmental performance, and a series of robustness checks supported the validity of the finding. Moreover, a heterogeneity analysis showed that this effect is more pronounced in non-heavily polluting industries, non-labor-intensive industries, and non-state-owned firms. In addition, the mechanism test showed that both enterprise innovation and digital transformation can positively moderate the inverted-U-shaped relationship between enterprise social insurance contribution burden and environmental performance, i.e., a reasonable and appropriate social insurance contribution burden can promote the green development of firms through enterprise innovation and digital transformation. Finally, under the appropriate proportion of social insurance contributions, firms with higher environmental performance can obtain higher enterprise value. This study extends the academic literature on the factors influencing corporate environmental performance from the perspective of corporate social insurance contributions, and provides empirical evidence for the government reform of social insurance policies and corporate adjustment of social insurance contribution rates to promote corporate sustainability.

The policy implications of this study are as follows. First, given that the aging of the population has become increasingly prominent, it is an urgent mission for the Chinese government to continuously improve the social security system and meet people's growing needs for a better life. At the same time, firms' green development cannot be ignored considering the worsening of environmental problems. Therefore, relevant policies should be formulated with a full consideration of the trade-off between social security and the sustainable business operation of firms, so as to avoid the negative impact of excessive corporate social insurance contribution burden on the green development of firms. Second, corporate innovation and digital transformation play important roles in dealing with the burden of social insurance contributions and improving corporate environmental performance. Due to the "double-edged sword" effect of corporate innovation and digital transformation, firms should actively seek deep integration of corporate green development with long-term development strategies including green innovation and digital transformation at a low level of social insurance contribution burden, and empower corporate green development through innovation and digital transformation. In addition, when the government and other management departments take measures to reduce the social security burden of firms such as social security fee reduction and tax cut, they should also guide firms to actively carry out green innovation and digital transformation, optimize internal governance mechanisms, and realize the transformation from human cost to human capital on the basis of scientific assessment of social security contribution rates. Third, we further improved the targeting and flexibility of social insurance premium adjustment policy measures. The inverted-U-shaped impact of the burden of corporate social insurance contributions on environmental performance varies significantly among firms with different characteristics, placing higher demands on the government to improve social insurance policies. It is necessary to guide substantial benefit measures such as "reducing taxes and fees" and "reducing costs" to firms with different characteristics at different stages, amplify the efficacy of macro policies in supporting the green development of real firms, and further enhance the precision, flexibility, and effectiveness of social insurance adjustment policies.

**Author Contributions:** Conceptualization, N.-C.W. and D.-K.S.; methodology, D.-K.S. and N.-C.W.; software, N.-C.W.; validation, N.-C.W.; data curation, N.-C.W.; writing—original draft preparation, N.-C.W. and C.-F.D.; writing—review and editing, N.-C.W., D.-K.S. and C.-F.D.; visualization, N.-C.W.; supervision, N.-C.W.; project administration, D.-K.S.; funding acquisition, D.-K.S. All authors have read and agreed to the published version of the manuscript.

**Funding:** This research was funded by Taishan Sholar Foundation of Shandong Province, grant number tsqn202103054.

**Institutional Review Board Statement:** Not applicable.

**Informed Consent Statement:** Not applicable.

**Data Availability Statement:** The data were mainly obtained from the CSMAR and the RESSET database.

**Conflicts of Interest:** The authors declare no conflict of interest.

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
