# Peer review of "Social Insurance Burden and Corporate Environmental Performance: Evidence from China"

_sustainability, doi:10.3390/su141912104_

Round 1

Reviewer 1 Report

Thank you for the opportunity to revise the paper ‘Social insurance burden and corporate environmental performance: Evidence from China’ submitted to Sustainability. Based on panel data, authors undertake the analysis of the relationship between social insurance and Chinese firms’ environmental performance. The research is well detailed, and I think that almost theoretical and methodological dimensions have been covered. There are just two points I would like to discuss before recommending acceptance: (i) what explains R2 low scores; and (ii) it would be great to detail this study results while comparing with the results from other studies. Good luck with your research!

Author Response

Responses to the Comments of the paper entitled “Social insurance burden and corporate environmental performance: Evidence from China” submitted to Sustainability(No.1886995).

First of all, we would like to thanks for your valuable and guiding comments. We appreciate your efforts in reviewing our study and your suggestions to improve the paper. We have paid high attention to each of your comments and tried to do our best to revise this paper.

We bold in blue our responses to the suggestions and comments of the editor and referees, and highlight all the changes made to the text in red.

Response to the referee’s comments:

#Comments

  1. Thank you for the opportunity to revise the paper ‘Social insurance burden and corporate environmental performance: Evidence from China’ submitted to Sustainability. Based on panel data, authors undertake the analysis of the relationship between social insurance and Chinese firms’ environmental performance. The research is well detailed, and I think that almost theoretical and methodological dimensions have been covered. There are just two points I would like to discuss before recommending acceptance: (i) what explains R2 low scores.

Response to the comments: Thanks so much for your recommendation. We really appreciate for your valuable suggestions and comments.

In terms of low R2, it is very common in panel data models. R2 represents the variance between unobserved and observed parts. Therefore, if we pay attention to the relationship between dependent variable and independent variables, R2 statistic does not explain the causality between dependent variable and independent variable. In other words, lower R2 statistic does not meaning smaller explanatory power of the model. It is worth noting that, in general, R2 of a panel-data model is lower than that of the time-series model, because there is strong heterogeneity in the panel data. In fact, we usually pay less attention to the values of R2, but more attention to the significance and sign of core explanatory variables. In order to avoid similar problems for readers and reviewers, we have cited some published paper in the revised version of the paper to support our model regression. Please see red area in this section.

2.It would be great to detail this study results while comparing with the results from other studies. Good luck with your research!

Response to the comments: Thanks very much for your suggestion and comment. Following your suggestion, we have provided more discussions of the empirical results by comparing our findings with those in other studies. Specifically, we have explained the differences of findings between our study and previous studies. In addition, we have also refined the marginal contribution of our paper and detailed the research innovation. Please see red area in the revised manuscript.

Finally, we would like to express our gratitude to you for the extremely helpful comments and for your guidance in the revision. We hope that our efforts have succeeded in allaying your concerns. We look forward to learning about your decision. And we express our thanks again to the anonymous referee for his time and efforts in reviewing our paper. Any remaining errors are our own.

Reviewer 2 Report

I think the paper is interesting and have a potential to become cited. However, I believe the explanatory poser could be improved by justifying the link between independent and dependent variable. I believe the authors could say innovation and particularly innovativeness is often a controlled process (Pesämaa, 2017). Such elaboration on control may strengthen the relationship to theory between performance measures and how you direct various activities. 

I would also suggest authors to add a recent reference on endogeneity test (see Pesämaa et al., 2021). 

Reference

Pesämaa, O. (2017). Personnel-and action control in gazelle companies in Sweden. Journal of Management Control28(1), 107-132.

Pesämaa, O., Zwikael, O., HairJr, J., & Huemann, M. (2021). Publishing quantitative papers with rigor and transparency. International Journal of Project Management39(3), 217-222.

Author Response

Responses to the Comments of the paper entitled “Social insurance burden and corporate environmental performance: Evidence from China” submitted to Sustainability (No.1886995).
First of all, we would like to thanks for your valuable and guiding comments. We appreciate your efforts in reviewing our study and your suggestions to improve the paper. We have paid high attention to each of your comments and tried to do our best to revise this paper.
We bold in blue our responses to the suggestions and comments of the editor and referees, and highlight all the changes made to the text in red.

Response to the referee’s comments:
#Comments
1.I think the paper is interesting and have a potential to become cited. However, I believe the explanatory poser could be improved by justifying the link between independent and dependent variable. I believe the authors could say innovation and particularly innovativeness is often a controlled process (Pesämaa, 2017). Such elaboration on control may strengthen the relationship to theory between performance measures and how you direct various activities. 
Response to the comments: Thanks very much for your suggestion and comment. In order to strengthen the causality relationship between independent and dependent variable, we have revised the paper from the following three aspects. First, in the model setting, we have controlled Firm fixed effect, and Year-Industry fixed effect to control external environment. Second, we have added some macro-level control variables to avoid potential omitting variables and potential endogenous problems. Third, we have added some robust tests by replacing dependent variable, independent variable to maintain the robustness of our main results. Please see red area in this section.
2. I would also suggest authors to add a recent reference on endogeneity test (see Pesämaa et al., 2021). 
Response to the comments: Thanks very much for your suggestion and comment. Following your guidance, we cite the papers of Pesämaa (2017, JMC) and Pesämaa et al.(2021, IJPM) to strengthen the relationship between dependent variable and independent variables. Please see red area in the revised manuscript.

Finally, we would like to express our gratitude to you for the extremely helpful comments and for your guidance in the revision. We hope that our efforts have succeeded in allaying your concerns. We look forward to learning about your decision. And we express our thanks again to the anonymous referee for his time and efforts in reviewing our paper. Any remaining errors are our own.

Reviewer 3 Report

This paper is an interesting study. From the perspective of paying social insurance for employees, this paper studies the impact of social insurance burden on enterprise environmental performance, and discusses the heterogeneity of different types of enterprises. On this basis, it further examines the moderating effect of innovation and digital transformation in the impact of social insurance burden on enterprise environmental performance. The logic of the article is complete and the empirical evidence is sufficient. The following two points are expected to be improved by the authors.

1) The review of previous studies in this paper focuses on the actual impact of social insurance burden on enterprise labor employment and wages, productivity and cash holdings and does not involve research related to enterprise environmental protection or environmental performance. The possible reason is that there are too few such studies. It is suggested to increase the content of literature review from other relevant perspectives. For example, since this paper thinks that the social insurance burden can actually be regarded as the labor cost of enterprises, how does labor cost affect enterprise environmental performance? Relevant literature should be supplemented.

2) It is suggested to highlight the contribution of the article in the conclusion.

Author Response

Responses to the Comments of the paper entitled “Social insurance burden and corporate environmental performance: Evidence from China” submitted to Sustainability (No.1886995).

First of all, we would like to thanks for your valuable and guiding comments. We appreciate your efforts in reviewing our study and your suggestions to improve the paper. We have paid high attention to each of your comments and tried to do our best to revise this paper.

We bold in blue our responses to the suggestions and comments of the editor and referees, and highlight all the changes made to the text in red.

Response to the referee’s comments:

#Comments

  1. This paper is an interesting study. From the perspective of paying social insurance for employees, this paper studies the impact of social insurance burden on enterprise environmental performance, and discusses the heterogeneity of different types of enterprises. On this basis, it further examines the moderating effect of innovation and digital transformation in the impact of social insurance burden on enterprise environmental performance. The logic of the article is complete and the empirical evidence is sufficient. The following two points are expected to be improved by the authors.

(1) The review of previous studies in this paper focuses on the actual impact of social insurance burden on enterprise labor employment and wages, productivity and cash holdings and does not involve research related to enterprise environmental protection or environmental performance. The possible reason is that there are too few such studies. It is suggested to increase the content of literature review from other relevant perspectives. For example, since this paper thinks that the social insurance burden can actually be regarded as the labor cost of enterprises, how does labor cost affect enterprise environmental performance? Relevant literature should be supplemented.

Response to the comments: Thanks very much for your suggestion and comment. Following your suggestion, we have added some relevant references to support the theoretical relationship between social burden and environmental protection. On the one hand, in the hypothesis development section, we have strengthened the discussions of how social insurance burden affects enterprise environmental performance; particularly, we have revised the discussions from two parts. First, we have explained how social insurance burden affects the cost of enterprise, and second, we have further analyzed how the labor cost of enterprise affect the environmental performance. On the other hand, we have added the economic meanings for the empirical results. In addition, we have also carried out heterogeneity analysis in order to provide auxiliary empirical evidence for the mechanism analysis. Finally, our extensive analysis provides more profound enlightenment for further understanding the microeconomic effect of enterprises’ social insurance burden. Please see red area in this section.

  1. It is suggested to highlight the contribution of the article in the conclusion.

Response to the comments: Thanks so much for your suggestion and comments. We have followed your suggestion to compare our findings with those of other studies. Specifically, by focusing on the differences of the findings between our study and the existing studies, we have refined the marginal contribution of our paper. Please see red area in the revised manuscript.

Finally, we would like to express our gratitude to you for the extremely helpful comments and for your guidance in the revision. We hope that our efforts have succeeded in allaying your concerns. We look forward to learning about your decision. And we express our thanks again to the anonymous referee for his time and efforts in reviewing our paper. Any remaining errors are our own.
